# Reproducibility study of "LICO: Explainable Models with Language-Image Consistency"

**Luan Fletcher**[*]                                      *luan.fletcher@student.uva.nl*
*Department of Computer Science*
*University of Amsterdam*

**Robert van der Klis**[*]                               *robert.van.der.klis@student.uva.nl*
*Department of Computer Science*
*University of Amsterdam*

**Martin Sedláček**[*]                                   *martin.sedlacek@student.uva.nl*
*Department of Computer Science*
*University of Amsterdam*

**Stefan Vasilev**[*]                                     *stefan.vasilev@student.uva.nl*
*Department of Computer Science*
*University of Amsterdam*

**Christos Athanasiadis**                                *c.athanasiadis@uva.nl*
*Department of Computer Science*
*University of Amsterdam*

*https: // github. com/ robertdvdk/ lico-fact*

## Abstract

The growing reproducibility crisis in machine learning has brought forward a need for careful examination of research findings. This paper investigates the claims made by Lei et al. (2023) regarding their proposed method, LICO, for enhancing post-hoc interpretability techniques and improving image classification performance. LICO leverages natural language supervision from a vision-language model to enrich feature representations and guide the learning process. We conduct a comprehensive reproducibility study, employing (Wide) ResNets and established interpretability methods like Grad-CAM and RISE. We were mostly unable to reproduce the authors' results. In particular, we did not find that LICO consistently led to improved classification performance or improvements in quantitative and qualitative measures of interpretability. Thus, our findings highlight the importance of rigorous evaluation and transparent reporting in interpretability research.

## 1   Introduction

Machine learning is facing a reproducibility crisis (Kapoor & Narayanan, 2022), necessitating the replication and validation of research findings. Our focus in this paper is on reproducing the main results and investigating claims from the work of Lei et al. (2023) regarding their proposed LICO method.

LICO aims to improve *model interpretability*: an active area of research, mainly because of its applicability in explaining model decisions. Especially in high-stakes industries such as healthcare, poor interpretability is a major blocker for wider adoption of deep learning methods (Zhang et al., 2020; Piccialli et al., 2021).

---

[*]Equal contribution

To better understand where LICO positions itself in this domain, we can first subdivide model interpretability approaches into two categories: *active* and *passive* (Zhang et al., 2020). *Active* methods either modify the architecture (i.e. *model-based* approaches (Murdoch et al., 2019)) or the training procedure (e.g. LICO) to achieve better interpretability, while *passive* or *post-hoc* approaches (used interchangeably) explain predictions of already trained models without modifications. Post-hoc methods assume that the features of the learned model are non-interpretable. By contrast, for model-based approaches the model itself or its parts can intrinsically help explain the prediction (e.g. weights in linear regression). We can further subdivide post-hoc interpretability methods into two main approaches — gradient-backpropagation-based (Smilkov et al., 2017; Sundararajan et al., 2017), where explainable noisy gradients are generated, and class activation mapping (CAM) based (Selvaraju et al., 2016; Chattopadhay et al., 2018; Wang et al., 2020), which produce saliency maps. Within this categorisation, LICO is seen as an active method that modifies the model training procedure leading to improved post-hoc class activation maps. LICO does not modify model architecture, unlike active model-based methods, making it generally compatible with current approaches in deep learning according to Lei et al. (2023).

Ordinarily, CAMs are generated using one-hot class labels to determine which pixels in the input image are important for the classification. However using one-hot encoded labels leads to biased and poor-quality latent representations (Vyas et al., 2020), because one-hot labels carry minimal semantic information. The insight provided by Lei et al. (2023) is that training a network for classification while using natural language supervision from a vision-language model such as CLIP (Radford et al., 2021) to regularise the feature maps may lead to more interpretable feature maps, as well as better classification performance. LICO aims to achieve this by shaping the latent feature space to approximate a semantically rich text encoder space via learnable prompts (Lei et al., 2023).

To investigate the claims about LICO from the work of Lei et al. (2023), we make quantitative and qualitative comparisons between ResNets/Wide ResNets (He et al., 2015) (Zagoruyko & Komodakis, 2017) with and without the modified training objective from LICO for a subset of the interpretability methods used in the original experiment — namely GradCAM (Selvaraju et al., 2016), GradCAM++ (Chattopadhay et al., 2018), and RISE (Petsiuk et al., 2018). We also measure classification accuracy for some of the datasets used in the original setup — CIFAR-10, CIFAR-100 (Krizhevsky et al., 2009), and subsets of ImageNet (Deng et al., 2009; Howard, 2019) to investigate the claims of improved classification performance.

In the remainder of our work, we first provide a more detailed outline of the specific claims made by Lei et al. (2023) about LICO (Section 2). Next, we explain how we investigated these claims (Section 3) and provide the results of our experiments in Section 4. Finally, in Section 5, we discuss the claims of the paper in the context of these results, and how reproducible the authors' results were.

## 2 Scope of reproducibility

The LICO method is claimed by Lei et al. (2023) to be a qualitative and quantitative enhancement of existing interpretability methods in deep learning. From their experiments and conclusions, we identified the following main claims of the paper:

1. *Training an image classifier with LICO consistently improves quantitative and qualitative model interpretability.*

2. *Training an image classifier with LICO consistently improves classification accuracy.*

3. *Tasks with more classification categories benefit more from LICO because the prompt guidance comes from a manifold consisting of more classes.*

Ultimately, this report aims to investigate the reproducibility of the LICO paper by attempting to fully or partially reproduce its main experiments and determine how strongly they back the aforementioned claims. In Section 3 we provide more detail on how the original authors set up their method and experiments and we compare their results directly with ours in Section 4.

## 3 Methodology

There are two training objectives added by LICO: firstly, the image manifold is encouraged to have a similar shape as the text prompt manifold through the manifold matching (MM) loss, and secondly, prompt tokens are mapped to specific feature maps and vice versa through the optimal transport (OT) loss. The MM loss coarsely aligns the manifold structure, whereas the OT loss establishes fine-grained relations between individual tokens and feature maps.

As a starting point for our implementation, we used the publicly available code by the authors Lei et al. (2023). As their codebase was incomplete, we implemented the training loop, the MM loss and the Insertion/Deletion metrics ourselves (see Section 3.4). Additionally, we made use of existing implementations of CAM-based interpretation methods (Gildenblat & contributors, 2021) and RISE (Ishikawa, 2019) to produce saliency maps.

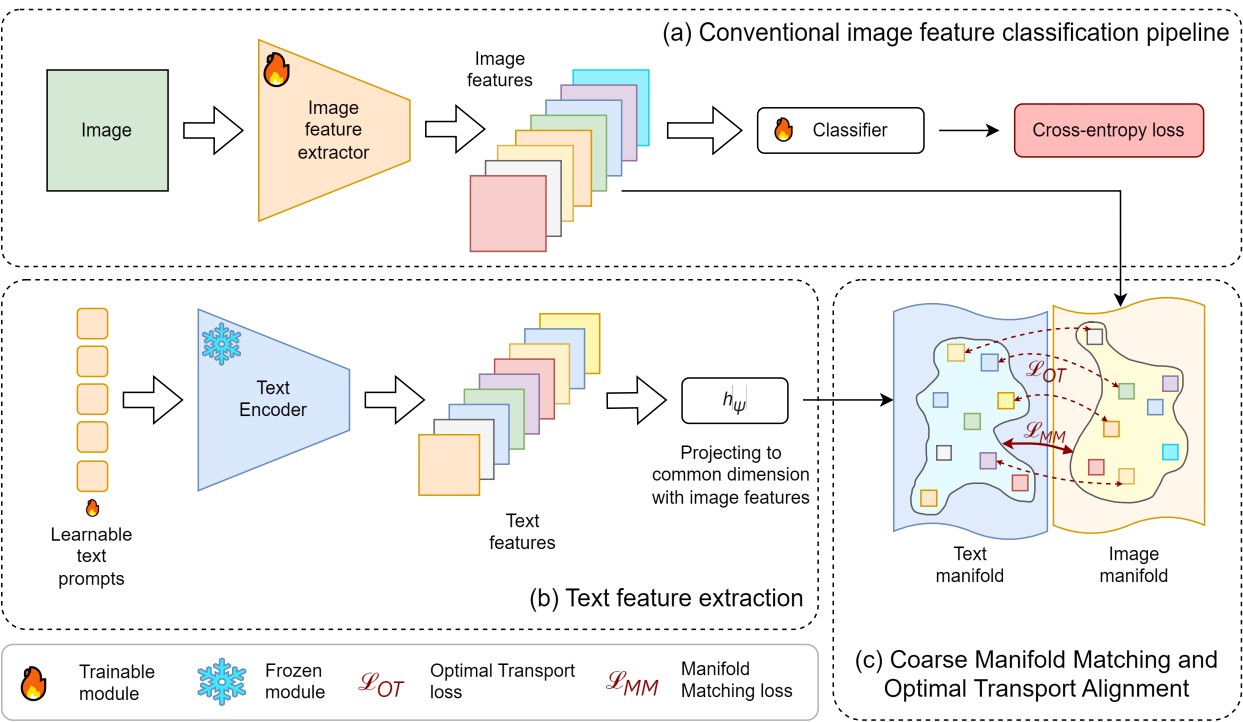

Figure 1: The pipeline of LICO Lei et al. (2023) (a) A regular image classification neural network pipeline. (b) Extraction of language features from learnable prompts and a frozen pre-trained text encoder. (c) A visualisation of the two objectives proposed in LICO: manifold matching and feature alignment by optimal transport.

### 3.1 Model descriptions

LICO consists of a vision-language model (VLM) with an image encoder, whose embedding space is structured by a pre-trained text encoder that has learned a robust semantic space. This "structuring" is achieved by adding the MM loss and the OT loss terms to the main cross-entropy loss objective.

The outputs of the image encoder $\boldsymbol{F} \in \mathbb{R}^{N \times d'}$ are $N$ feature map embeddings of dimension $d'$ (Figure 1a). The inputs to the text encoder $\boldsymbol{t}_i = [X_1, X_2, \ldots, X_{M-1}, t_i]$ are learnable prompt context tokens $X_m$ concatenated with $t_i$, where $t_i$ is the text from the class label (Figure 1b). Each class has its own prompt input $\boldsymbol{t}_i$ with $i \in \{1, ..., K\}$, where $K$ is the number of classes. The output of the text encoder for a given $\boldsymbol{t}_i$ is $\boldsymbol{G}_i \in \mathbb{R}^{M \times d}$. Since the dimensions of the image feature maps and the text features $d'$ and $d$ are of different sizes, the authors use an extra mapping multi-layer perceptron $h_\psi[d, d']$ to map the text features to the dimension of the image features, as seen in Figure 1b.

The MM loss aims to match the manifolds of image features and text features of corresponding samples in a batch in embedding space by treating them as two multivariate distributions and minimizing KL divergence between them. To transform them to distributions, the authors use the matrices $\boldsymbol{A}_{B \times B}^G$ and $\boldsymbol{A}_{B \times B}^F$ corresponding to $\boldsymbol{G}$ and $\boldsymbol{F}$, defined as

$$\boldsymbol{A}_{i,j}^F = \frac{\exp\left(-D\left(\boldsymbol{F}_i, \boldsymbol{F}_j\right)/\tau\right)}{\sum_{s=1}^{B} \exp\left(-D\left(\boldsymbol{F}_i, \boldsymbol{F}_s\right)/\tau\right)}, \quad \boldsymbol{A}_{i,j}^G = \frac{\exp\left(-D\left(\boldsymbol{G}_i, \boldsymbol{G}_j\right)/\tau\right)}{\sum_{s=1}^{B} \exp\left(-D\left(\boldsymbol{G}_i, \boldsymbol{G}_s\right)/\tau\right)}, \tag{1}$$

where $D(.,.)$ is a distance metric, like Euclidean distance, and $i, j$ index specific feature map embeddings within a training batch of size $B$. Here, Lei et al. (2023) state that $\tau$ is a learnable parameter. We will cover the implications of this design choice in Section 3.3.

While the MM loss provides a coarse alignment of the structure of the feature maps and encoded prompts, the OT loss aims to match individual feature maps in $\boldsymbol{F}_i$ to individual tokens in $\boldsymbol{G}_i$ by solving the optimal transport problem (Monge, 1781), thus providing a fine-grained mapping between the two sets of embeddings. Since the problem is intractable in continuous space, it is discretized through the Sinkhorn algorithm (Cuturi, 2013). To transform the image and text features $\boldsymbol{F_i}, \boldsymbol{G_i}$ to discrete distributions, Lei et al. (2023) first perform $L_2$ normalization over the feature dimension. Then, the normalized feature space is divided into $N$ and $M$ bins respectively via Dirac functions placed at the centre of each bin, which is the discretization step. Formally, $\boldsymbol{\mu}$ and $\boldsymbol{v}$ are the discretized distributions corresponding to $\boldsymbol{F_i}$ and $\boldsymbol{G_i}$:

$$\boldsymbol{\mu} = \sum_{n=1}^{N} u_n \delta_{\boldsymbol{f}_n}, \quad \boldsymbol{v} = \sum_{m=1}^{M} v_m \delta_{\boldsymbol{g}_m}, \tag{2}$$

where $\delta_{\boldsymbol{f}_n}$ and $\delta_{\boldsymbol{g}_m}$ are the Dirac delta functions placed at the centre of each bin and $u_n$ and $v_m$ are weights in the probability simplex.

In practice, these are set to uniform weights. Then, via the Sinkhorn algorithm, which speeds up the computation of the optimal transport problem by orders of magnitude, as compared to OT solvers, a transport matrix $T$ and a cost matrix $C$ are obtained. These are multiplied element-wise and summed to get the total optimal transport cost (OT loss):

$$\mathcal{L}_{\text{OT}} = \frac{1}{B} \sum_{b=1}^{B} \mathbf{1}^T (C_b * T_b) \mathbf{1} \tag{3}$$

The MM loss is coarse because it aggregates the set of embeddings of each input sample, be it image or text, *to a single vector representation*, transforms it to a distribution and then uses this per-sample representation to calculate a KL divergence with other sample representations. The optimal transport objective, on the other hand, is argued to be fine-grained because it does not aggregate the feature maps of a sample to a single representation vector but *transforms each feature embedding in $\boldsymbol{F}_i$ and $\boldsymbol{G}_i$ into respective discrete distributions, whose respective bins represent single embeddings in $\boldsymbol{F}_i$ and $\boldsymbol{G}_i$.*

The total loss of LICO is then $\mathcal{L} = \mathcal{L}_{\text{CE}} + \alpha \mathcal{L}_{\text{MM}} + \beta \mathcal{L}_{\text{OT}}$.

### 3.2 Datasets

We evaluate LICO's classification performance and measure Insertion/Deletion scores (Petsiuk et al., 2018; Zhang et al., 2021; Wang et al., 2020), as defined in Section 3.4, on CIFAR-10, CIFAR-100 (Krizhevsky et al., 2009), Imagenet (Deng et al., 2009), and Imagenette (Howard, 2019) datasets. The CIFAR datasets are well-established in literature, while Imagenette is a subset of 10 classes of the ubiquitous ImageNet dataset with 160×160 pixel images. We turn to this alternative because the images in it are larger, like in Imagenet, which might mean that they stand to benefit more from LICO in the interpretability evaluation due to the saliency maps being more detailed. At the same time, running on Imagenette instead of Imagenet keeps the amount of compute necessary for multiple training runs manageable. To investigate the results of LICO on the original dataset that the authors used, we also complete a single training run on Imagenet with all 1000 classes but *20% of the data.*

Even though the authors use CIFAR-10/100 only to measure classification performance, we perform further investigation and additionally evaluate Insertion/Deletion scores. For both datasets, we split the dataset into 47,500 training, 2,500 validation and 10,000 test samples. On CIFAR-100, we also create a subset consisting of only 2,500 examples to compare the performance of a baseline model vs. a model trained with LICO when training with limited data.

For Imagenette, we assess both classification accuracy and Insertion/Deletion scores (Section 3.4). The dataset is divided into 9000 training samples, 475 validation samples, and 3925 test samples.

During training, we use two augmentations: random horizontal flip and random crop of size $H \times W$ with a reflect padding of 4, where $H$ and $W$ are the image height and width, respectively. For both training and test sets, we also perform normalization with per-channel mean/std statistics.

We also evaluate the LICO model on the PartImageNet dataset (He et al., 2022). This dataset has segmentation masks so that we can evaluate the Intersection over Union (IoU) of the generated saliency maps with the segmentation masks. This provides us with another metric of saliency map quality, as a high score means that the network is consistently looking at exclusively the salient parts of images. Since these are ImageNet images, we use the standard ImageNet augmentations: random horizontal flip, resize the short side to 224, random crop, and normalize with per-channel mean/std statistics. In the evaluation phase, we skip the horizontal flip and exchange the random crop for a center crop.

### 3.3 Hyperparameters

In our reproduction we mainly aim to adhere to the implementation details of the LICO paper. For more details, readers are referred to Section A in the Appendix. The Wide ResNet we use for training on CIFAR-10 has depth factor 28 and widening factor of 2, as this was also used by Zhang et al. (2022); Sohn et al. (2020), and the authors of LICO used many of the experimental settings from those papers. On CIFAR-100 we use a depth of 28 and a widening factor of 8 for the same reason, and on Imagenette we use the same settings as on CIFAR-100. The number of context tokens in a prompt is set to 12, as the ablation study from Lei et al. (2023) shows that the LICO losses give the best results with 12 context tokens.

We also take a closer look into the authors' design choice of $\tau$ (Equation 1) being a learnable parameter. It should be noted that as $\tau$ increases to arbitrarily large values, both $\mathbf{A}_{i,j}^F$ and $\mathbf{A}_{i,j}^G$ approach a constant value:

$$\lim_{\tau \to \infty} \frac{\exp(-D(a,b)/\tau)}{\sum_{s=1}^{B} \exp(-D(a,b)/\tau)} = \frac{\exp(0)}{\sum_{s=1}^{B} \exp(0)} = \frac{1}{B}.$$

Thus, in the limit, both $\mathbf{A}^F$ and $\mathbf{A}^G$ will be matrices with all entries equal to $\frac{1}{B}$. This, in turn, means that

$$\mathcal{L}_{\mathrm{MM}} = \frac{1}{B} \sum_{i=1}^{B} \mathrm{KL}[\mathbf{A}_{i,:}^G || \mathbf{A}_{i,:}^F] = 0.$$

Thus, it seems likely that $\tau$ should either not be learnable, or if it is learnable, it ought to be clipped to a maximum value. We perform experiments with it being either a learnable parameter or a constant $\tau = 1$ to determine whether this matters in practice.

### 3.4 Experimental setup and code

In this study, we ran experiments aiming to reproduce the main parts of the LICO paper and its results. Our code and hyperparameters follow exactly the methodology and values described in the paper, but we run two experiments beyond the original paper: one with the aim to investigate the generalisability of the authors' claims about model interpretability by evaluating the IoU between ground-truth segmentation masks and saliency maps. With the other auxiliary experiment, we aimed to investigate whether the $\tau$ parameter should be a learnable parameter or a constant.

As image encoders, we use Wide ResNet for the CIFAR-10, CIFAR-100, and Imagenette datasets, and we use a ResNet18 (He et al., 2015) for the Imagenet and PartImageNet datasets.

Three metrics were used to measure the performance of LICO against a non-LICO trained model — classification accuracy, Insertion/Deletion scores, and the IoU between generated saliency maps and ground-truth segmentation masks. The Insertion score is determined by starting from a blurred input image, and gradually unblurring pixels, where the order in which pixels are unblurred is determined by the saliency map. Then, if the saliency map has correctly assessed which pixels are important for the network to make its prediction on the partially blurred image, the class probability should quickly rise as a function of unblurred pixels. The Insertion score is the AUC of this curve, so a good Insertion score is close to 1. The Deletion score is measured analogously, by starting with the base image, and gradually removing pixels; a good Deletion score is close to 0. We measure Insertion/Deletion in conjunction with interpretation methods — Grad-CAM, Grad-CAM++ and RISE. This follows the same setup as the original experiment conducted by Lei et al. (2023). We also show an Overall score (i.e. insertion - deletion) in conjunction with the Insertion/Deletion like the authors of LICO.

As for the IoU scores, they serve to evaluate whether the model accurately focuses on relevant areas within the image. A high IoU indicates consistent attention to the object of interest rather than the background. To incorporate IoU into our analysis, we must interpret it within the context of non-binary saliency maps and binary ground truth labels as regions for comparison. We utilize the original IoU formula, which is sensible because we can view the saliency map region as a soft assignment of a region for segmentation. This approach offers an alternative metric for assessing saliency map quality, leveraging segmentation masks as a potentially stronger signal than class probabilities in insertion/deletion scores. Furthermore, we hypothesize that since segmentation maps directly highlight the object of classification, they serve as an effective proxy for evaluating saliency map quality.

Lastly, our code is available on GitHub[*].

## 4 Results

We have performed extensive experimentation with the aim of testing the claims of the authors. It includes testing the interpretability claims and the classification accuracy improvement claims of the authors. We further show our results on an ablation study on the different loss terms introduced and on our extensions of the paper's experiments — comparing segmentation labels with our saliency maps and the effects of a learnable $\tau$ parameter.

Overall, our results do not support the central claims of the paper. Saliency map interpretability did not increase either quantitatively or qualitatively for models trained using LICO. In addition, the use of LICO decreased classification performance across the board. The results of our extension are also in lin with these findings.

In the following subsections, we will analyze the results of each experiment in detail and provide evidence for our discussion in the following section.

---

[*] https://github.com/robertdvdk/lico-fact

### 4.1 Claim 1: Improvement in interpretability

Table 1: Baseline/LICO Insertion, Deletion, and Overall (Insertion - Deletion) scores for CIFAR-10, CIFAR-100, Imagenette, and ImageNet (20%). Numbers in bold are the highest for a specific interpretability method.

| Dataset | Method | Grad-CAM++ | Grad-CAM | RISE |
|---|---|---|---|---|
| CIFAR-10 | Insertion | **53.8**/48.7 | 63.0/**63.6** | **73.3**/72.4 |
| | Deletion | **43.8**/46.9 | 37.3/**35.4** | **31.8**/30.0 |
| | Overall | **9.9**/1.7 | 25.8/**28.2** | 41.5/**42.4** |
| CIFAR-100 | Insertion | **34.1**/33.8 | **44.0**/43.4 | **54.3**/54.2 |
| | Deletion | **27.6**/28.6 | **19.2**/19.8 | **13.4**/14.3 |
| | Overall | **6.5**/4.2 | **26.2**/23.6 | **40.9**/39.9 |
| Imagenette | Insertion | **60.2**/59.2 | **77.1**/74.5 | **70.9**/70.3 |
| | Deletion | **36.9**/38.3 | **23.0**/25.5 | 36.2/**35.8** |
| | Overall | **23.3**/21.0 | **54.1**/49.0 | **34.7**/34.5 |
| ImageNet | Insertion | **29.9**/23.8 | **32.6**/24.7 | - |
| | Deletion | **7.5**/5.0 | **5.3**/4.4 | - |
| | Overall | **22.4**/18.8 | **27.3**/20.4 | - |

As shown in the results in Table 1, we did not replicate the authors' findings that LICO generally improves the interpretability of saliency maps. With the exception of CIFAR-10 using Grad-CAM/RISE, LICO reduced our quantitative measure of interpretability (Insertion/Deletion scores) for every saliency map method and dataset tested.

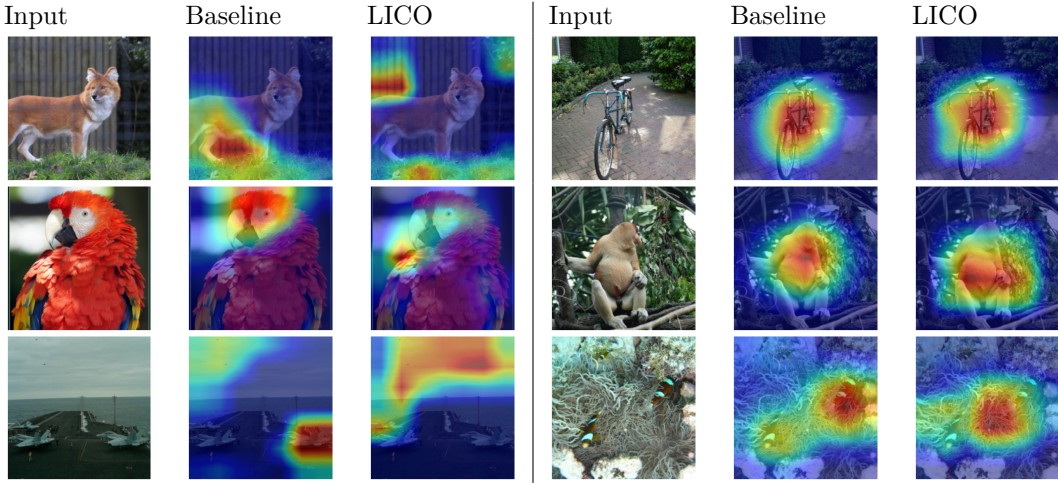

Figure 2: Examples of GradCAM saliency maps for a model trained with and without LICO. Both models were trained on the same 20% subset of ImageNet. On the left, we see the validation set images that the LICO authors used, while on the right we see images from the test set, picked by us.

As for the qualitative side of measuring interpretability, Figure 2 also demonstrates that the generated saliency maps do not show any obvious qualitative improvement when training with LICO, counter to the authors' claims. Furthermore, we argue that LICO does not visibly produce more explainable attention maps than the baseline: neither in our saliency maps, nor in those of the original paper.

On the left in Figure 2 we see the original images from the validation set that the authors use. We get different saliency maps than the authors, possible because we have fewer images per class. Apart from that, we note that LICO's saliency maps seem to be focused slightly less on important regions of the images than

the baseline. This hints towards worse generalization ability for a dataset with many classes and few images per class. We show in Section 4.3.1 that worse saliency maps also translate to lower accuracy than baseline.

## 4.2 Claims 2 and 3: Improvement in classification accuracy

We find that LICO slightly decreases test accuracy on all four datasets (Table 2), as well as on the smaller subset of CIFAR-100. These results also contradict the authors' claim that a greater number of classification categories leads to a greater increase in classification performance when using LICO: we find that LICO does not perform any better in the ImageNet and CIFAR-100 cases with 1000 and 100 classes respectively, than in the CIFAR-10 and Imagenette cases with 10 classes.

Similarly, our results also contradict the authors' assertion that LICO is particularly beneficial in circumstances with limited data. We instead find that in the limited data case, the degradation in performance due to LICO is significantly worse than in the full data case. It should be noted that, in general, our results are much worse than the authors' results in this case. This leads us to believe that the authors did something that was not described in the paper to improve results on limited data. This missing part might also explain why, in our case, LICO was much worse with limited data, whereas in their case, LICO was better.

Table 2: Test accuracy of baseline models against LICO models on ImageNet, Imagenette, and CIFAR10/100. The ImageNet model was trained once, the other models were trained three times with different seeds. The ImageNet baseline model was ResNet-50, and for every other dataset we used a Wide ResNet-18. A limited case with 2500 examples for CIFAR-100 is also evaluated.

|          | ImageNet | Imagenette | CIFAR-10 | CIFAR-100 | CIFAR-100 (2500 examples) |
|----------|----------|------------|----------|-----------|---------------------------|
| Baseline | **49.75** | **87.3** $\pm$ 0.74 | **93.9** $\pm$ 0.13 | **78.1** $\pm$ 0.66 | **28.5** $\pm$ 0.39 |
| +LICO    | 48.18    | 86.8 $\pm$ 1.58 | 93.6 $\pm$ 0.29 | 77.3 $\pm$ 0.90 | 18.6 $\pm$ 1.64 |

## 4.3 Loss ablation study

As a further investigation into why most of our results differ from the authors, we performed an ablation study on the individual loss components. Specifically, we trained models on CIFAR-10, CIFAR-100 and Imagenette using only cross entropy loss and one of the OT/MM losses.

### 4.3.1 Classification performance

Table 3: Test accuracy of LICO with ablated losses (over three runs) on CIFAR-10/100, Imagenette and CIFAR-100 in the limited data setting.

|                | Imagenette | CIFAR-10 | CIFAR-100 | CIFAR-100 (2500 examples) |
|----------------|------------|----------|-----------|---------------------------|
| Baseline       | 87.3 $\pm$ 0.74 | 93.9 $\pm$ 0.13 | 77.7 $\pm$ 0.08 | **28.5** $\pm$ 0.39 |
| OT only        | 83.8 $\pm$ 1.58 | 93.6 $\pm$ 0.35 | 78.0 $\pm$ 0.38 | 23.6 $\pm$ 1.96 |
| MM only        | **88.0** $\pm$ 0.26 | **93.9** $\pm$ 0.11 | **78.6** $\pm$ 0.33 | 26.1 $\pm$ 1.10 |
| LICO (OT + MM) | 86.8 $\pm$ 1.58 | 93.6 $\pm$ 0.29 | 77.0 $\pm$ 1.08 | 18.6 $\pm$ 1.64 |

In the original paper, the authors perform one ablation study on the individual losses only for the Imagenet dataset. It shows that using only the OT loss leads to a degradation in test accuracy versus baseline and using only the manifold loss leads to an improvement in test accuracy versus baseline. We find that training with the OT loss alone leads to a small drop or no change in test accuracy while training with the manifold loss alone leads to a small increase or no change in test accuracy, approximately replicating the authors' results (Table 3).

The one exception to our findings is in the case of limited data, where the baseline model remains significantly better than all other models. Even here, however, the model trained using only the manifold loss performs the best of the three non-baseline models.

The authors also find that using both losses leads to a greater improvement in test accuracy than using the manifold loss alone. We do not see this in our results. Instead, we find that using both losses is always worse than using the manifold loss alone.

### 4.3.2 Interpretability

In the authors' ablation study, both the OT loss by itself and the manifold loss by itself improve the interpretability of saliency maps as measured by Overall scores, and they find that the best performance on this metric is achieved with the use of both losses. Our results do not bear this out, with the baseline models more often than not performing the best (Table 4). We also do not see a clear pattern of one loss providing more improvement than another, with all of the non-baseline models performing approximately equally when averaged over the different datasets and CAM methods.

Table 4: Ablation study results of LICO with MM/OT losses evaluated for interpretability: Insertion, Deletion and Overall scores. The results are on CIFAR10/100 and Imagenette datasets.

| Dataset | Method | GradCAM | | | GradCAM++ | | | RISE | | |
|---|---|---|---|---|---|---|---|---|---|---|
| | | Ins | Del | Ovr | Ins | Del | Ovr | Ins | Del | Ovr |
| CIFAR10 | Baseline | 63.0 | 37.3 | 25.4 | **53.8** | **43.8** | **9.9** | 73.3 | 31.8 | 41.5 |
| | MM only | 59.3 | 39.5 | 19.8 | 51.5 | 46.4 | 2.6 | 72.8 | 30.9 | 41.9 |
| | OT only | 63.0 | 36.8 | 26.2 | 53.5 | 44.4 | 9.1 | **73.9** | 30.4 | **43.5** |
| | LICO (OT + MM) | **63.6** | **35.4** | **28.2** | 48.7 | 46.9 | 1.7 | 72.4 | **30.0** | 42.4 |
| CIFAR100 | Baseline | **44.0** | **19.2** | **26.2** | **34.1** | 27.6 | **6.5** | 54.3 | 13.4 | 40.9 |
| | MM only | 41.0 | 21.2 | 19.8 | 31.8 | 29.2 | 2.6 | 55.0 | **13.3** | 41.7 |
| | OT only | 41.3 | 20.8 | 20.5 | 32.9 | **27.5** | 5.4 | **55.1** | 13.3 | 41.8 |
| | LICO (OT + MM) | 43.4 | 19.8 | 23.6 | 32.8 | 28.6 | 4.2 | 54.2 | 14.3 | 39.9 |
| Imagenette | Baseline | **77.1** | **23.0** | **54.1** | 60.2 | **36.9** | **23.3** | **70.9** | 36.2 | **34.7** |
| | MM only | 75.9 | 27.0 | 48.9 | **60.4** | 39.0 | 21.4 | 69.1 | 38.0 | 31.1 |
| | OT only | 71.7 | 27.3 | 44.4 | 57.3 | 38.3 | 19.0 | 64.7 | 37.8 | 26.9 |
| | LICO (OT + MM) | 74.5 | 25.5 | 49.0 | 59.2 | 38.3 | 21.0 | 70.3 | **35.8** | 34.5 |

## 4.4 Results beyond original paper

### 4.4.1 Saliency maps and segmentation ground truths comparison

We perform a further investigation of the model's interpretability by evaluating the IoU between ground truth segmentation masks and saliency maps generated with Grad-CAM (5). Over three runs, we get a mean IoU of 0.185 for the baseline model against 0.141 for LICO, which is 23.7% lower. This is a substantial difference, suggesting that LICO produces worse-quality saliency maps because it focuses less on the segmented object of classification and is possibly more spread out. As a visual example, in Figure 2, Example 2 we see that the saliency magnitude is low even though it is focused on the correct object, which would result in a low IoU for this image.

| Method | Intersection over Union |
|---|---|
| Baseline | $0.185 \pm 0.005$ |
| LICO | $0.141 \pm 0.008$ |

Table 5: Intersection over Union scores, evaluated on PartImageNet

### 4.4.2 The effect of a learnable versus constant $\tau$

To investigate the effect of using a fixed or learned temperature, we performed an experiment on CIFAR-10 where we compared 3 models: one without LICO losses, one with LICO losses but with fixed temperature, and one with LICO losses and with learned temperature.

| Method | Accuracy | GradCAM | | | GradCAM++ | | | RISE | | |
|---|---|---|---|---|---|---|---|---|---|---|
| | | Ins | Del | Ovr | Ins | Del | Ovr | Ins | Del | Ovr |
| Baseline | **93.6 ± 0.09** | 63.9 | 38.0 | 25.8 | **56.0** | **44.1** | **11.8** | **69.7** | **34.0** | **35.7** |
| LICO ($\tau$ learnable) | 93.5 ± 0.08 | **66.8** | 36.6 | **30.1** | 50.1 | 47.9 | 2.2 | 68.5 | 34.5 | 34.0 |
| LICO ($\tau$ fixed) | 93.4 ± 0.08 | 65.2 | **36.5** | 28.7 | 49.2 | 49.2 | 0.0 | 67.7 | 36.2 | 31.4 |

Table 6: Test accuracy, Insertion/Deletion and Overall scores for the experiment with $\tau$ for LICO. The setups are LICO with learnable $\tau$ against fixed $\tau$ with a baseline no LICO training.

In Table 6 we see that keeping $\tau$ fixed does not seem to have any beneficial effect. The test accuracy is not significantly different from the test accuracy with a learnable $\tau$, and the Insertion/Deletion scores are generally worse than with a learnable $\tau$. One possible reason for the lack of significant differences in our results is that the LICO losses do not provide any benefit. Consequently, parameters that adjust how the LICO losses work might have no effect if the LICO losses themselves are ineffective.

Our hypothesis with this experiment was that keeping $\tau$ fixed would prevent the manifold loss from vanishing during training, therefore positively influencing the training process. However, it does not have a significant effect across datasets. Therefore, the design choice of $\tau$ being learnable is unlikely to matter for the results obtained from LICO.

## 5 Discussion

In general, our results did not support the claims of the paper. We extensively tested the methodology of the paper across CIFAR-10 and CIFAR-100 datasets, because these datasets were manageable in size while also having results in the paper we could compare to. We performed many iterations and variations of experiments in an effort to replicate the reported results. However, in all our experiments, we did not observe consistent improvements in accuracy, nor did we detect qualitative or quantitative enhancements in interpretability.

We did replicate some of the findings of the authors regarding ablation on losses. Specifically, we found that using the manifold-matching loss alone leads to consistently better classification performance than using the OT loss alone. However, we did not replicate the finding that using both losses performs best. This implies that much of the difficulty in replicating the authors' findings is due to issues with the OT loss. It is possible that our implementation of the OT loss differs substantially from that of the authors, but we are unable to confirm this due to incomplete code provided by them and a lack of response to our inquiries seeking clarification. We started experimenting with the OT loss code provided by the authors in their repository, but this, too, did not yield the expected results reported by Lei et al. (2023) in the original paper. Finally, we attempted to implement the OT loss ourselves from scratch, exactly as it is described in the LICO paper. This implementation also failed to reproduce the reported results despite this being the intended implementation of the OT loss, based on its description by Lei et al. (2023) (see section 5.2 for further details).

A limitation of our study is that our implementation may differ substantially from the implementation the authors used to generate their results (see Section 5.2). It is likely that we missed several setup details and optimizations that the authors used to get the networks to train slightly better, resulting in a slightly better accuracy. Thus, all we can conclude is that the paper is not reproducible as it is now, given the existing resources: the paper itself, and the public code repository.

### 5.1 What was easy

The paper gives both an intuitive as well as a formal description of the losses, and pseudocode for the training algorithm, which is very helpful in understanding how the method works and why. Thus, the core contributions of the paper are well explained.

In addition, the paper provides extensive experimentation and results with discussions to solidify the scope of the proposed methods and to evaluate them from different aspects. We found this very helpful for understanding why and to what extent LICO performs well.

### 5.2 What was difficult

Attempting to reproduce the LICO paper was, in general, quite difficult. This was caused by three main factors: firstly, the code [*] provided by the authors was incomplete. The MM loss function and the training loop were missing, as well as the implementations for a ResNet50 (He et al., 2015) and PARN (He et al., 2016) with prompts. We attempt to alleviate this problem by providing our own codebase, where many parts of the methodology are implemented exactly as described by Lei et al. (2023).

Secondly, in several cases, the provided code did not match the methodology as described in the paper. Most notably, the paper proposed "to align $G_i$ (...) and $F_i$ (...) for achieving consistency between feature maps and specific prompt tokens". However, the implementation in the code provided by Lei et al. (2023) reduces the output of the text encoder into a single feature vector per class instead. This does not correspond to the description in the paper, and based on our observations, this discrepancy renders the OT loss meaningless. This reduction to a single vector in the authors' implementation is realized via an "." (EOT) token contained in the corresponding prompt $G_i$ for every class, which is being used like a [CLS] token in ViT (Dosovitskiy et al., 2021). Additionally, in the paper, the learned context is shared between the prompts for every class: in $G_i$, the only part that depends on the sample index is $g_{t_i}$. In the code, there is a separate learned context for every class prompt.

Finally, details of the experimental setup were ambiguous, which exacerbated the difficulty of reproduction. The paper did not describe the depth and width of the WideResNet used to generate their results on CIFAR-10, CIFAR-100, and SVHN. Instead, they referred to another paper's setup without pointing out what part of the experimental setup was used. The paper also missed a justification for a key design choice: in Equation 1, $\tau$ is defined to be a learnable parameter, which leads to issues the MM loss (see Section 3.3 for a discussion of the implications). This prompted our investigation in Section 4.4.2.

### 5.3 Communication with original authors

We attempted to establish communication with the authors persistently over a span of more than four weeks. Regrettably, we received no response to any emails. This substantially hindered the efforts to reproduce the original experiments, as the code provided was incomplete and inconsistent with the paper itself on multiple occasions (see Section 5.2 for more detail). We were only provided the code in this state after emailing the authors about the lack of code in the paper's repository. This was the only time we received any response to our queries, although the emails themselves never received an explicit reply.

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

## A  Hyperparameters

All hyperparameters are shown in Table 7.

We keep $\alpha = 10$ and $\beta = 1$ for all experiments. The batch size is kept at 64 for all setups other than ImageNet, and the learning rate at 0.03. We use SGD as optimizer with momentum 0.9 and weight decay 0.0001.

We apply the cosine learning rate scheduler $\eta = \eta_0 \cos\left(\frac{7\pi k}{16K}\right)$ from the paper (Lei et al., 2023), and we train on CIFAR-10/100 for 200 epochs and on ImageNet and Imagenette for 90 epochs.

| Hyperparameter | Value |
|---|---|
| $\alpha$ | 10 |
| $\beta$ | 1 |
| Batch size | 64 and 128 (ImageNet) |
| Learning rate | 0.03 |
| Optimizer | SGD |
| Momentum | 0.9 |
| Weight decay | 0.0001 |
| Learning rate scheduler | Cosine |
| Learning rate scheduler parameters | $\eta = \eta_0 \cos\left(\frac{7\pi k}{16K}\right)$ |
| Training epochs (CIFAR-10/100) | 200 |
| Training epochs (Imagenette, ImageNet) | 90 |

Table 7: Hyperparameters for experiments

## B  Computational requirements and environmental impact

For our runs we used NVIDIA A100 GPUs on a cluster. Completing a single training run on CIFAR-10 with and without LICO took around 2.5 hours, on CIFAR-100 around 4 hours, on Imagenette approximately 1.5 hours, and on PartImageNet around 2 hours.

To calculate the environmental impact of our reproducibility study, we use the following formula:

$$\mathrm{CO_2}e = \mathrm{CI} \cdot \mathrm{PUE} \cdot \mathrm{E},$$

where CI is Carbon Intensity (g $\mathrm{CO_2}e$/kWh), PUE is the Power Usage Effectiveness (no unit), and E is the energy used (kWh). For the CI, we use the most recent value provided by the European Environment Agency: in the Netherlands in 2022, generating one kWh of electricity emitted an amount of greenhouse gases equivalent to 321 grams of $\mathrm{CO_2}$ [*]. As for the PUE, we do not know the value of Snellius specifically, but the European Commission has stated that the average value in the EU is 1.6 [*], so that is the value we will use. In our efforts to reproduce the paper, we used an estimated 205 kWh (calculated using the "eacct" command of the "Energy Aware Runtime (EAR) SLURM plug-in"). Putting this all together, we arrive at a total of

$$321\mathrm{g\ CO_2}e/\mathrm{kWh} \cdot 1.6 \cdot 205\mathrm{kWh} \approx 105\mathrm{\ kg\ CO_2}e.$$

If we compare this to the greenhouse gas emissions per capita, which was 9600 kg $\mathrm{CO_2}e$ per year in the Netherlands in 2021 [*], we find that running the experiments for our paper used approx. 11% of what a Dutch citizen uses per year. Alternatively, comparing to the EU $\mathrm{CO_2}$ emission target for passenger cars, which is 95 g $\mathrm{CO_2}$/km, our study used the same amount of $\mathrm{CO_2}e$ as driving a passenger car for approx. 1105 km.

---

[*] `https://www.eea.europa.eu/en/analysis/indicators/greenhouse-gas-emission-intensity-of-1`

[*] `https://joint-research-centre.ec.europa.eu/jrc-news-and-updates/eu-code-conduct-data-centres-towards-more-innovative-sustai`
`en`

[*] `https://climate.ec.europa.eu/eu-action/transport/road-transport-reducing-co2-emissions-vehicles/`
`co2-emission-performance-standards-cars-and-vans_en`

## C    Hyperparameter search

In order to investigate the impact of the hyperparameters on performance, we trained a model on CIFAR-100 with varying hyperparameters and a set seed. Below we report these results. The best results in each table are bolded.

### C.1    Weight of manifold and OT losses

Table 8: CIFAR-100 test accuracy by manifold loss weight ($\alpha$) and OT loss weight ($\beta$).

|  |  | $\beta$ | | | | |
|---|---|---|---|---|---|---|
|  |  | 0 | 1 | 2 | 5 | 10 |
|  | 0 | 77.9 | 77.6 | 76.5 | 75.4 | 77.6 |
|  | 1 | **78.7** | 78.2 | 76.3 | 76.0 | 77.0* |
| $\alpha$ | 2 | 78.2 | 77.8 | 77.2 | 76.6 | 75.4 |
|  | 5 | 77.8 | 77.8 | 77.4 | 76.6 | 76.5 |
|  | 10 | **78.7** | 77.0 | 76.1 | 76.8 | 74.4 |

There is a general trend of decreasing performance as the weight of the OT loss is increased. Further, the best performance is achieved when the OT loss is not used at all (i.e. when $\beta = 0$).

### C.2    Weight decay

Table 9: CIFAR-100 test accuracy by weight decay.

| Weight Decay | $10^{-2}$ | $10^{-3}$ | $10^{-4}$ | $10^{-5}$ | $10^{-6}$ |
|---|---|---|---|---|---|
| Val Accuracy | 19.7 | 73.8 | **77.0***  | 76.7 | 76.0 |

The weight decay value used in the original paper appears to lead to the best performance.

### C.3    Number of epochs trained

Table 10: CIFAR-100 test accuracy by numbers of epochs trained.

| Number of Epochs | 200 | 300 | 500 |
|---|---|---|---|
| Test Accuracy | **77.0*** | **77.0** | 76.3 |

There does not appear to be any improvement in accuracy when increasing the number of epochs trained for.

* These are the hyperparameters used in the LICO paper

