# OpenReview forum: "Reproducibility study of “LICO: Explainable Models with Language-Image Consistency""
_TMLR — Accepted by TMLR_

### Review · Reviewer_SYve · 2024-03-14

**Summary Of Contributions:**

- This paper attempts to reproduce the key claims and experiments from the original LICO paper, which proposed using natural language guidance from vision-language models to improve image classification performance and interpretability.
- The authors re-implemented the LICO method and losses (manifold matching and optimal transport) and evaluated it on CIFAR-10, CIFAR-100, Imagenette and PartImageNet datasets.
- They measured classification accuracy as well as quantitative metrics like insertion/deletion scores and IoU with segmentation masks to assess interpretability improvements claimed by the original paper.

**Audience:**

Yes

**Claims And Evidence:**

Yes

**Requested Changes:**

Overall I think this is a very decent effort to reproduce LICO.
I believe the efforts the authors put into trying to reproduce this paper have been reasonable.
I guess one could argue that more effort to find the missing hyperparameters and prompts could have made the reproducibility more successful.

Could you please articulate how much effort went into trying to do the above? Apologies If I have missed that bit in our paper.

**Strengths And Weaknesses:**

The author(s) of this paper did a very good job of trying to reproduce the claims made in the original paper.

In terms of the strengths:
- Thorough evaluation across multiple datasets and interpretation methods to comprehensively test the original claims (they used CAM-based interpretation methods proposed by Gildenblat et al., 2021 and RISE (Ishikawa, 2019).
- Implemented additional experiments beyond the original paper, like IoU computation (due to a dataset used that has segmentation masks  available) and ablation on loss terms.
- Identified issues in the original paper's description and provided code that complements that of the original authors (incomplete hence hindering reproducibility).
- Transparent reporting of computational requirements and environmental impact (author of the reproducibility paper added an appendix related to this).

Weaknesses:
I do not think the below-listed weaknesses are necessarily attached to the reproducibility effort but some limitations are:

- Unable to successfully reproduce most of the original paper's findings, especially on improved interpretability and classification accuracy with LICO (lack of source code definitely an issue but it is not clear whether the effort without the source code, i.e. just by replicating the paper, fell short or the details in the original paper were not enough; I am not certain).
- Had to make several implementation choices due to ambiguities and missing details in the original paper (again similar to point 1 above).
- Lack of response from original authors to clarify issues that did not help with the reproducibility efforts (especially given that some information on loss functions is missing from the original paper)

---

> ### Author Response · Authors · 2024-04-08
>
> We thank the reviewer for their time and effort in reviewing our submission.
>
> Thank you for pointing out the lack of clarity regarding hyperparamter search. We did perform quite a bit of hyperparameter search in our attempts to implement the paper, trying many variations of weights on the OT loss and manifold matching loss in terms of the $\alpha$ and $\beta$ hyperparameters, the weight decay parameter, and the number of epochs trained for. We have now put our results from that search into the appendix.
>
> In our results, use of the OT loss generally resulted in worsened performance (as measured by test accuracy), while using the manifold matching loss alone slightly improved performance. This is consistent with our claim in the discussion section that the issues with the original paper mostly lie with the OT loss.
>
> In order to investigate whether our models might not be converging, we tested a number of training epochs, up to 500 (2.5x the number of epochs used in the original paper). We did not see any further convergence when compared with the number of epochs used in our main experiments.
>
> Finally, we tried varying the weight decay parameter, and did not find any improvement in results when deviating from the value in the paper.
>
> Hopefully, this answers your question and we would be happy to provide further clarification if needed.

---

> > ### Comment · Reviewer_SYve · 2024-04-12
> > **thank you**
> >
> > Many thanks for the responses.
> > I am happy with the explanations provided.
> > As the other reviewers have mentioned, a run on Imagenet could have provided some additional evidence to substantiate your claims, but in the grand scheme of things and lack of resources, I believe Imagenette is a good proxy.

---

### Review · Reviewer_C77H · 2024-03-24

**Summary Of Contributions:**

This paper investigates the reproducibility of LICO [1]. The authors make the following key contributions:

1. The authors provide a clear review of the methods and claims in LICO.
2. Since the original LICO repository lacked information necessary for reproducing the original experimental results, the authors have reimplemented the LICO code and made it publicly available.
3. The authors evaluate the effectiveness of LICO by demonstrating its impact on both interpretability and classification accuracy.

[1] Yiming Lei et al. LICO: Explainable Models with Language-Image Consistency. NeurIPS 2023.

**Audience:**

No

**Broader Impact Concerns:**

Please refer to the weaknesses.

**Claims And Evidence:**

Yes

**Requested Changes:**

- [introduction] What is different between model-based approaches and class activation mapping based approaches? Is there no overlap between the two methods? Categorizing methods with paper citations would improve the readability of the paper.
- [3.2 dataset] Imagenette and ImageNet contain 14k images and over 1M images, respectively. Considering Imagenette as a large dataset may be an overstatement.
- [4.1 Results] It would be valuable to use the same images in LICO, for the qualitative experiments. The left image in Figure 2 does not provide useful insights since the baseline algorithms (GradGAM) has already poor results.
- [A Hyperparameters] In LICO, the batch size is 128 for ImageNet experiments. It would be valuable if the Imagenette experiments are also conducted with the same batch size, not 64.

Minors
- [introduction] It would be more appropriate to place the citation at the correct location. For example, changing `between ResNets and Wide ResNets with and without the LICO losses (He et al., 2015; Zagoruyko & Komodakis, 2017)` into `between ResNets (He et al., 2015; Zagoruyko) and Wide ResNets (Zagoruyko & Komodakis, 2017) with and without the LICO losses`

**Strengths And Weaknesses:**

### Strengths
- The paper is well-structured.
- The authors clearly highlight how difficult it is to reproduce the LICO paper. This underscores the importance of a thorough peer review process.
- Experimental results are well presented, where the authors employ diverse datasets (Imagenette (small subset of ImageNet), CIFAR-10, and CIFIAR-100) and baseline algorithms (GradCAM, GradCAM++, and RISE).

### Weaknesses
- The current manuscript could benefit from experiments with larger datasets such as ImageNet and more complex models such as ResNet-50. Although the current experiments offer valuable insights, it is worth mentioning that that the key comparison on interpretability and classification performance in the original LICO paper were conducted with ImageNet and ResNet-50. Not only will a larger dataset allow the model to capture more precise saliency maps, but a larger model may also be useful for learning substantial information in the CLIP language encoder (VIT-B/32).
- A more in-depth ablation study of the OT loss function used in LICO is required. It is acknowledged that precisely implementing the OT loss is challenging, as the authors mentioned the Discussion section. However, the paper does not sufficiently show an attempt to implement the OT loss. Interestingly, experiments in Table3 shows the efficacy of the MM loss of LICO.
- The clarity of the writing could be improved in some sections. Specific suggestions are provided below.

---

> ### Author Response · Authors · 2024-04-08
>
> Thank you for the constructive review.
>
> > * [introduction] What is different between model-based approaches and class activation mapping based approaches? Is there no overlap between the two methods? Categorizing methods with paper citations would improve the readability of the paper.
>
> Thank you for the suggestion. We have edited the introduction section with a description of a categorisation of interpretability methods following a prominent survey work [1]. The categorisation divides interpretability methods into passive (post-hoc) and active ones (with both LICO and model-based approaches falling into the latter). We further expanded our description with a subdivisions of both active and post-hoc interpretability methods, citing relevant methods, making a clear distinction between the categories, and explaining how the active LICO can aid post-hoc methods. We hope this change makes the distinctions clear and helps the reader correctly position LICO in the landscape of interpretability methods.
>
> > * [3.2 dataset] Imagenette and ImageNet contain 14k images and over 1M images, respectively. Considering Imagenette as a large dataset may be an overstatement.
> > * [A Hyperparameters] In LICO, the batch size is 128 for ImageNet experiments. It would be valuable if the Imagenette experiments are also conducted with the same batch size, not 64.
>
> We would like to highlight that we never claim that Imagenette is a large-scale dataset, but rather we evaluate on it because of the larger image size, because “we theorize that testing on this dataset might be important because larger image sizes might benefit more in the interpretability evaluation due to possibly more detailed saliency maps”. However, we agree with the remark and recognize that testing on ImageNet is desirable for verifying the authors claims. We now added an extra training setup on a 20% subset of the Imagenet dataset, where a 20% subset of each class in the train split is taken. While we lack the computational resources to run on the full Imagenet dataset, we move a step forward into testing the original claim of the authors and testing their method on a larger-scale dataset. We also ran the training with batch size 128, as it is in the original paper and as you suggested.
>
> > * [4.1 Results] It would be valuable to use the same images in LICO, for the qualitative experiments. The left image in Figure 2 does not provide useful insights since the baseline algorithms (GradGAM) has already poor results.
>
> We succeeded in locating the exact images that the authors of LICO use for their saliency maps experiment, even though no marker was provided to make finding them easy. We also updated Figure 2 to display those images (which the original authors picked to be part of the validation set) and three others from the test set, chosen by us.
>
> Hopefully, these changes have answered your concerns and we would be happy to provide further clarification if needed.
>
>
> [1] Zhang, Yu, et al. "A survey on neural network interpretability." IEEE Transactions on Emerging Topics in Computational Intelligence 5.5 (2021): 726-742.

---

> > ### Comment · Reviewer_C77H · 2024-04-13
> > **Thank you**
> >
> > Thank you for the additional experiments and paper revisions., these address my concerns and improve the paper. I am pleased that in section 5, the authors acknowledge the difficulty of implementing OT loss effectively. Nonetheless, an ablation study of the performance changes for the hyperparameters of OT loss would be beneficial.

---

### Review · Reviewer_A9LE · 2024-03-25

**Summary Of Contributions:**

The paper attempts to reproduce the main findings in the LICO paper. The authors first summarize the main claims in that paper, then describe the method in the original paper as well as the experimental setup for their reproducibility study and, finally, they report and discuss their results. The authors find that none of the claims in LICO holds for their reproducibility study.

**Audience:**

Yes

**Broader Impact Concerns:**

No concerns.

**Claims And Evidence:**

Yes

**Requested Changes:**

* Conduct the analysis with ImageNet.
* Rewrite Section 3 with original figures and an interpretation of the original LICO paper, rather than copying content from that paper.
* Polish the text and provide insights as to why the paper could not be reproduced.

**Strengths And Weaknesses:**

**Strengths:**

[+] Reasonable reproducibility study:

The authors have done a good job at summarizing the findings in LICO, defining an experimental setup to test these hypothesis, and showing that in their setup they were not able to reproduce the results in the original paper,

**Weaknesses:**

[-] The scope of the paper is quite limited:

The paper limits itself to reproducing another method (unsuccessfully). As such, it only provides evidence towards invalidating the claims in LICO. The study adds a small experiment checking the IoU of saliency and segmentation maps for a baseline and a model trained with LICO, but does little to go beyond the original paper and to provide insights as to why the study was unsuccessful.

[-] The experimental setup is limited compared to the original LICO paper.

While the authors tested some of the datasets in LICO, they mostly used small datasets (CIFAR-10/0). While it is true that the claims do not hold for such datasets, for thoroughness it would be important to test other setups (SVHN, ImageNet) in the paper.

[-] Errors/typos in the text, copied figures, repeated content.

The paper is quite unpolished, with a considerable amount of grammatical errors and typos/spelling mistakes that make it difficult to read it. Additionally, Figure 1 in the paper is a copy of Figure 2 in LICO and is not properly attributed. Finally, Section 3 contains mostly content already found in the LICO paper, with some of the formulas copied verbatim.

---

> ### Author Response · Authors · 2024-04-08
>
> Thank you for the constructive review.
>
> > * Conduct the analysis with ImageNet.
>
> We thank you for this suggestion and recognize the value of this experiment. We would have liked to train on ImageNet, however, due to computational constraints we could not complete the run on the full dataset. Instead, we trained on a subset of ImageNet, where a 20% subset of the training samples were kept for each class. We found that the claims of LICO still do not hold for this setup, which is a step further in our investigation, which is in line with our other results.
>
> > * Rewrite Section 3 with original figures and an interpretation of the original LICO paper, rather than copying content from that paper.
>
> We edited our methodology description in Section 3 by adding minor details in the MM loss description and crafting our own interpretation of the OT loss method. We also added an original Figure 1 visualising the LICO method in a (hopefully) more readable manner instead of copying it from the LICO paper.
>
> > * Polish the text and provide insights as to why the paper could not be reproduced.
>
> We attempted to polish the text and expanded on parts such as the efforts to replicate the OT loss and difficulties with the original codebase. Specifically in section 5, we discuss such issues in detail and hope to give the reader an explanation as to why our efforts in replicating the paper were unsuccessful.

---

### Decision · Action_Editor_GNVe · 2024-05-28

**Recommendation:** Accept as is

**Comment:**

The reviewers unanimously recommend the publication of the paper.  This represents an interesting study of reproducibility.  Beyond the investigation of the reproducibility of the LICO paper, this work will help future papers avoid pitfalls that could impede reproducibility.
 Hence I recommend a reproducibility certification.

**Audience:**

The paper will interest the community working on explainability and reproducibility.

**Claims And Evidence:**

The paper claims that the LICO paper is not reproducible.  The paper describes various attempts to reproduce the results while discussing the challenges it encountered.